# The Created Excellent Thermal, Mechanical and Fluorescent Properties by Doping Eu^3+^-Complex-Anchored Carbon Nanotubes in Polycyanate Resins

**DOI:** 10.3390/nano12122040

**Published:** 2022-06-14

**Authors:** Ziyao Hu, Dong Zhao, Yao Wang, Linjun Huang, Shichao Wang, Sui Mao, Olga Grigoryeva, Peter Strizhak, Alexander Fainleib, Jianguo Tang

**Affiliations:** 1Institute of Hybrid Materials, National Centre of International Joint Research for Hybrid Materials Technology, National Base of International Sci. & Tech. Cooperation on Hybrid Materials, Qingdao University, 308 Ningxia Road, Qingdao 266071, China; brokenangelzyao@163.com (Z.H.); zhaodong9712@163.com (D.Z.); wangyaoqdu@126.com (Y.W.); huanglinjun@qdu.edu.cn (L.H.); wangsc@qdu.edu.cn (S.W.); maosui001@qdu.edu.cn (S.M.); 2Institute of Macromolecular Chemistry, National Academy of Sciences of Ukraine, 02155 Kyiv, Ukraine; grigoryevaolga@i.ua (O.G.); fainleib@i.ua (A.F.); 3L. V. Pysarzhevsky Institute of Physical Chemistry, National Academy of Sciences of Ukraine, 31 Prosp. Nauky, 03028 Kyiv, Ukraine; pstrizhak@hotmail.com

**Keywords:** cyanate ester resin, carbon nanotubes, Eu complex, mechanical properties, fluorescence

## Abstract

In the blending process of the composites, the clustering of MWCNTs under high concentration leads to poor dispersion and difficult complexing with luminescent elements. Cyanate ester resins (CERs) have a brittle network structure when cured caused by a conjugation effect that forms a strong emission peak in the ultraviolet-visible region and quenches the luminescent elements of the fluorescent nanofillers. In this paper, by anchoring of the Eu complex (Eu(TTA)_3_Phen, ETP) on a surface of longitudinal split unzipped carbon nanotubes (uMWCNTs); fluorescent nanoparticles were prepared as ETP anchor unzipper carbon nanotubes (ETP-uCNTs). Dicyanate ester of bisphenol E (CER-E monomer) is cured to polycyanurate at a lower temperature to achieve a high conversion, promoting a uniform blend with ETP-uCNTs, providing the fluorescence environment with high color purity. Studies show the ETP-uCNTs solve the agglomeration of MWCNTs and improve the interface binding ability. Compared with the pure CER-E, the tensile strength, bending strength and impact strength of CER-E/0.8 wt.% ETP-uCNT hybrid nanocomposites are increased by 94.6%, 92.8% and 101.1%, respectively. The carbon residue rate of CER-E/ETP-uCNTs is up to 47.14% at 800 °C, the temperature of the maximum reaction rate decreases by 67.81 °C, and the partial absorption of ultraviolet light is realized between 200 and 400 nm.

## 1. Introduction

Cyanate ester resins (CERs) are a new type of heat-resisting thermosetting resin with excellent molding processing, dimensional stability, anti-wrinkle crack resistance, low moisture absorption and excellent dielectric properties for application in high-speed-communication electronic equipment in aeronautics and astronautics, missile materials, high-temperature encapsulation materials and automotive components manufacturing [1,2,3]. CER is a series of compounds containing the cyanate functional group (-OCN), mainly bisphenol dicyanate ester with structural formula NCO-R-OCN; the R group usually consists of aromatic rings [4,5]. CERs are polymerized via a cyclotrimerization reaction at high temperature in the presence of specific catalysts or without them, and the generated active intermediates have catalytic effects as well [6]. The uncatalyzed polycyclotrimerization requires higher curing temperature and longer curing time and the conversion rate without a catalyst is lower [7]. The densely crosslinked polycyanurate spatial network formed from CER has a highly symmetrical structure, but it is very rigid and brittle, which makes the tensile, bending and impact strength of the composites unsatisfactory and limits its application. In this regard, catalyzing the curing reaction of monomers, reducing the curing end-temperature and improving the performance without conventional catalysts are key steps. On the other hand, based on maintaining high thermostability, solving the brittleness problem is another research focus of composites. In addition, the development of multifunctional composites is also a new research direction and hotspot.

Multiwalled carbon nanotubes (MWCNTs) have high toughness and electrical properties; however, due to its smooth tube walls, it is difficult to anchor the lanthanide fluorescent complex, and it is difficult to band it with a polymer. Newly introduced unzipped multiwalled carbon nanotubes (uMWCNTs) are hollow structures obtained from MWCNTs [8,9]. The uMWCNTs retain a small number of original carbon atoms and have the characteristics of a thicker lattice layer. As a filler with high toughness, electrical and thermal properties, uMWCNTs and the polymer matrix can achieve excellent blending molding [10]. In order to prepare multifunctional and synergistic high-performance composites, we introduced the fluorescent complex to address another fatal defect of MWCNTs—the poor interfacial adhesion with resins resulting in poor mechanical properties and the appearance of bubbles. Through the step of unzipping, the external radius of MWCNTs is increased and many oxygen-containing active groups are introduced on the surface. The large specific surfaces and bonding connections enhance the anchoring of the luminescent elements and enhance the interfacial bonding ability of the matrix; this is due to the curing process of the monomer catalyzed by the nanoparticles. The lanthanide fluorescent complex has strong ultraviolet absorption in the ultraviolet-visible region. In recent years, it has often been used as a new linear material for ultraviolet protection and red light under ultraviolet radiation [11]. We have chosen ETP because it produces high fluorescence and quantum yield [12], and forms two ligands, 1,10-phenanthroline (Phen) and 2-acyltrifluoroacetone (TTA), [Eu(TTA)_3_Phen] (ETP), and emits red fluorescence due to the f-f transition of 4f electrons in Eu^3+^ [13]. In the previous work [14], we first prepared nanoparticles from ETP-anchored uMWCNTs, and then doped the nanoparticles with dicyanate of bisphenol A monomer (CER-A monomer) to prepare CER-A nanocomposites, endowed with mechanical properties and fluorescence properties of CER nanocomposites. However, the fluorescent intensity was weak, caused by the background fluorescence (430 nm) suppressed ETP (612 nm), the nanocomposites do not emit characteristic red fluorescence under excitation and the mechanical properties were lower within the limit range of low concentrations (0.01 wt.%).

In this work, the low viscosity of dicyanate ester of bisphenol E monomer (CER-E monomer) has a rotational flexibility structure. During the curing of monomers, this excellent structure enables it to achieve high conversion rates at lower temperatures. The uMWCNTs with lower electronegativity and ETP with higher activity were able to form coordination bonds, which prepare new derivative fluorescent nanoparticles (ETP-uCNTs), as shown in Figure 1. The ETP-uCNTs retains clearly detectable fluorescence emission when exposed to ultraviolet irradiation. The empty electron orbitals of ETP-uCNTs can form coordination bonds with N and O atoms in the polymer matrix to enhance the interface adhesion and the dispersion of ETP-uCNTs in the polymer matrix, which improves the toughness and reduces the nonradiative transition rate.

Thus, the CER-E/ETP-uCNT hybrid nanocomposites are prepared by blending CER-E monomer with ETP-uCNTs. Chemical bonding of ETP-uCNTs and CER-E monomer achieves effective dispersion at high concentration. The catalytic performance of ETP-uCNTs promotes a curing reaction, leading to the high toughness and excellent fluorescent environment of the nanocomposites; they can be used in aircraft casing piping and high-temperature industrial manufacturing applications. In addition, the multifunctionality of CER-E/ETP-uCNT fluorescent hybrid nanocomposites would have the sensitivity of a ^5^D_0_→^7^F_2_ transition to fluorescence emission intensity; they open a possibility for the development of new nanocomposites using special anti-counterfeiting materials, marking materials and protective materials. Finally, the nanocomposites have high strength, high UV protection and high sensitivity monitoring.

## 2. Materials and Methods

### 2.1. Materials

Cyanate ester resin (CER) based on dicyanate ester of bisphenol E was supplied by Kaixin New Material Technology Co., Ltd. (Hengyang City, China). Multi-walled carbon nanotubes were purchased from Xianfeng Nanomaterials Technology Co., Ltd. (Jiangsu, China). Potassium permanganate, sulfuric acid, hydrochloric acid, acetone, and hydrogen peroxide were purchased from Beijing National Pharmaceutical Group Chemical Reagent Co., Ltd. (Beijing, China). Europium (III) chloride and 2-Thenoyltrifluoroacetone (TTA) were purchased from Aladdin Industrial Corporation Shanghai, China. 1,10-Phenanthroline Phen was purchased from Sinopharm Chemical Reagent Co., Ltd. (Shanghai, China). All the reagents were used without further purification.

### 2.2. Experimental Section

#### 2.2.1. Modification of MWCNTs (uMWCNTs)

The preparation of uMWCNTs follows the improved Hummers method, also known as the oxidative decompression method [15]. A 0.3 wt.% H_2_O_2_ solution was prepared and frozen in advance. A 200 mg amount of MWCNTs was mixed with 35 mL of the concentrated H_2_SO_4_ solution and stirred uniformly mechanically for 10 min, and then ultrasonically for 10 h. Under the condition of strictly controlled temperature at about 20 °C, 40 mg KMnO_4_ was slowly added in five portions, with an interval of 10 min and stirred evenly to obtain a black dispersion. Then, the dispersion obtained stirred for 2 h at 50 °C to obtain a brown liquid. After that the brown liquid obtained was poured into ice cubes containing 0.3% H_2_O_2_ prepared in advance to eliminate the remaining KMnO_4_ and kept there until outgassing finished. Finally, the resulting liquid was centrifuged and washed until the pH value was about 4–5, and was freeze-dried and set aside, with the powder being obtained for later use.

#### 2.2.2. Preparation of ETP-MWCNTs and ETP-uCNTs

For preparation of ETP-MWCNTs/ETP-uCNTs, the dry powder obtained in Section 2.2.1 was used for chemical anchoring. The solutions of 0.1 mol/L Phen, 0.1 mol/L EuCl_3_, and 0.1 mol/L TTA were mixed in the required amounts and 1 mol/L dilute ammonia solution was added until pH = 6–7 was reached to prepare the Eu complex (Eu(TTA)_3_Phen, ETP).10 mg MWCNTs or uMWCNTs were added to 10 mL acetone solution for ultrasonic dispersion, and then the Eu(TTA)_3_Phen complex solution was added, stirred and ultrasonically treated. Finally, the mixed dispersion was centrifuged and washed, and the solid powder obtained was vacuum dried.

#### 2.2.3. Synthesis of CER/ ETP-MWCNT and CER/ ETP-uCNT Hybrid Nanocomposites

For preparation of ETP-MWCNT and CER/ ETP-uCNT composites, a certain amount (0.2 wt.%, 0.4 wt.%, 0.6 wt.%, 0.8 wt.%, 1.0 wt.%) of MWCNTs, uMWCNTs, ETP-MWCNTs or ETP-uCNTs was used. A 10 mg amount of nanoparticles was dispersed in 10 mL of acetone and stirred ultrasonically, and then added to the pre-polymerized CER, stirred to prepare a homogeneous mixture, and cast into a preheated tetrafluoroethylene mold coated with vacuum silicone grease. Next, the composition was degassed in a vacuum drying oven at 90 °C for 4 h to eliminates the solvent. Finally, the composition was polymerized according to the following curing schedule: 100 °C/3 h + 150 °C/2 h + 180 °C/2 h +210 °C/1 h + 230 °C/1 h.

### 2.3. Characterization

The morphological structures of unzipped carbon nanotubes were studied using the Transmission Electron Microscopy (TEM) method (JEM-1200EX, 100 kV), (TEM, JEM-F2100) (JEOJ, Kyoto, Japan). To characterize the crystal structure, the XRD (Bruker D8, Cu Kα radiation, λ = 1.5406 Å) (Bruker, Karlsruhe, Germany) technique was used to record the X-ray diffraction spectrum of powder solids in the scanning range of 10–80° (2θ). Raman spectroscopy system (Thermo Fisher Scientific, Waltham, MA, USA) was applied using 1.5 mW and 532 nm He-Ne laser as the excitation source. Fourier transform infrared spectroscopy with a MAGNAIR 550 infrared spectrometer was used to detect the vibration absorption peaks of the nanoparticles and the CER-based system (KBr pellets). XPS was conducted at Thermo Scientific K-Alpha (JEOJ, Ltd., Japan) to test the elemental composition and chemical bonding of the modified particles. STA 449F5 (TA Instruments, New Castle, PA, USA) was used to analyze the thermal stability of the composite in nitrogen from room temperature to 800 °C at 10 °C/min. The DSC technique (Germany NETZSCH, Cologne, Germany) was used to perform isothermal tests and non-isothermal tests in the range of 50 –400 °C (10 °C/min, q N_2_ = 50 mL/min). A universal testing machine (CMT4304GD) (MTS, Jinan, China) was used for tensile, flexural and impact tests. Dumbbell-shaped samples with a thickness of 4 mm, an inner and outer width of 5 and 10 mm and a length of 75 mm were placed on the lower plate, and the external force was applied at a constant rate of 2 mm/min (5–7 samples for each composition were tested). The Scanning Electron Microscopy (SEM) technique (SEM, JEOL 6460) (JEOJ, Kyoto, Japan) was used to analyze the cross-section of CER composites. UV-1700 ultraviolet spectrophotometer and a full-spectrum microscopic spectrophotometer (Craic 20/30, PerkinElmer, Waltham, MA, USA) were used to study the optical properties. The fluorescence spectra of composites were analyzed with FLS1000 (Edinburgh Instruments Ltd., Edinburgh, UK) for PL spectrum analyses.

## 3. Results

### 3.1. Morphology and Structure Analysis of ETP-uCNTs

Figure 1 shows the TEM images of the modified carbon nanotubes and the corresponding average diameter distribution histogram. In Figure 1a, the pure MWCNTs have hollow tubular structure, smooth surface, sharp edge and narrow diameter range. MWCNTs were obviously entwined, and high concentrations lead to agglomeration. [16]. Figure 1b shows the TEM image of the uMWCNTs; the hollow tube structure after strong oxidation was partially broken, and the average hollow tube diameters (see the peaks in the diameter distribution histograms) are increased from 23.92 ± 0.62 nm/43.8 ± 2.47 nm to 38.55 ± 1.80 nm/77.68 ± 4.55 nm, which shows the longitudinal width and the specific surface area are increased. The uMWCNTs with less folding, fewer layers and less aggregation were obtained, which correlates well with the observations of Mahmoud et al. [15]. After potassium permanganate is oxidized, amorphous uMWCNTs are formed around the graphite wall and the tubular structure is transformed into a similar graphene nanoribbon, indicating that the carbon grid structure is significantly damaged with the formation of many defects. This amorphous layer reduces an interfacial tension between the MWCNTs and the polymer matrix in the nanocomposite [8]. Figure 1c confirms that the doping of ETP does not destroy the original specific morphology of carbon nanotubes. The smooth wall of MWCNTs became rough, and the protruding part of the wall edge was ETP, indicating that ETP was successfully adsorbed on the wall surfaces of MWCNTs and uMWCNTs. Figure 1d clearly shows the structure is unzipped, the hollow tubular structure becomes blurred and the ETP is attached to the surface of the uMWCNTs. It is not clear if the ETP embedding is due to the large area ratio caused by unzipping, and only some small dark parts can be observed. ETP was anchored on the surface of uMWCNTs to form ETP-MWCNTs/ETP-uCNTs. The unzipped process can provide more space to adsorb ETP, which may improve the fluorescence intensity of composites.

Figure 2a shows the Raman spectra for the original MWCNTs and the uMWCNTs recorded with a 532 nm (2.34 eV) laser excitation wavelength and a characteristic D peak at 1357 cm^−1^, G peak at 1592 cm^−1^, 2D peak at 2710 cm^−1^ and 2G peak at 2956 cm^−1^. The D peak is caused by sp^3^ hybridization of MWCNTs at defect sites, and the G peak is caused by sp^2^ hybridization of well-ordered carbon atoms. The peak values of D and G reflect that the crystal structure of carbon nanotubes changes from ordered to defective. The 2D peak is caused by a strong signal that reacted in the regular structure after Raman double resonance scattering. The ratio of D peak to G peak intensities (R = I_D_/I_G_) represents the regularity of the structure and the graphitization degree of carbon nanotubes [9]. The R value of the uMWCNTs (1.09) is higher than the original MWCNTs (0.61). Compared with MWCNTs, the 2D peak of uMWCNTs is significantly reduced, which is caused by a broken lattice structure of uMWNTs. In the process of unzipping, oxygen-containing functional groups appear on a surface of the original MWCNTs, which increases surface defects and the disorder degree of MWCNTs. As a result, the increase of the R ratio can be considered as a confirmation of the successful modification of MWCNTs.

Figure 2b shows the XRD spectra of MWCNTs and uMWCNTs. The peak 002 (full width of a half peak near 26°, d002 = 3.4 Å) represents the interlayer spacing between adjacent graphene layers; the peak 004 indicates that the CNTs maintain a regular tube structure; the peak 100 (full width of a half peak near 43°) represents a graphite structure [9]. Compared to the characteristic peaks of the MWCNT spectra, the peaks of the uMWCNTs were not so sharp, which indicates a loss of the graphite structure of the carbon nanotubes. The peak 002 of uMWCNTs is slightly widened and displaced to a lower angle and the peak 004 disappears, indicating destruction of the tube structure. uMWCNTs were prepared by unzipping. The oxidation of KMnO_4_ destroys the complete lattice structure of carbon nanotubes and produces diffraction peaks of half peak width.

The XPS spectra (Figure 2d–f) show the element composition and distribution for MWCNTs, uMWCNTs, ETP-MWCNTs and ETP-uCNTs. The atomic percentage of each element is shown in Table 1. The O1s spectrum shows that uMWCNTs and ETP-uCNTs have higher oxygen content than MWCNTs and ETP-uMWCNTs, and the O/C increases, which confirms that the unzipping behavior increases oxygen-containing groups, as the peak at about 534.24 eV is attributed to the C=O group and about 531.88 eV is the C-O group. The ETP-MWCNTs and ETP-uCNTs have a peak near 532 eV corresponding to the Eu-O group, which is evidence of successful chemical grafting of the ETP to uMWCNTs. In the total spectrum (Figure 2d), the content of Eu indicates the attachment of the ETP to the surface of MWCNTs and uMWCNTs; bonds between surface carbon atoms are formed, causing the carbon nanotubes to be stretched and broken in the longitudinal direction. The peak at about 1135.71 eV of ETP-uCNTs for the Eu_3_d_5_ proves the bonding of Eu^3+^. Comparing the content of Eu (see Table 1), we can see that the ETP-uCNTs are more effective (Eu—1.76 %) than ETP-MWCNTs (Eu—0.84 %), which ensures the excellent fluorescence characteristics of the nanocomposites to be prepared.

Figure 2c shows the UV-Vis absorption spectra of MWCNTs, uMWCNTs, ETP, ETP-MWCNTs and ETP-uCNTs. The maximum UV absorption wavelength of MWCNTs and uMWCNTs was mainly in the range of 225–275 nm. The strong wide peaks are mainly C-C or C=C. The increased strength of uMWCNTs was due to the sp^2^ hybridization transition to the π-π* bond, confirming that uMWCNTs maintained a movement toward the visible region with reduced sp^2^ characteristics. The -C=O- shoulder peak of uMWCNTs appears near 335 nm due to the n-π* transition [9,17], corresponding to π → π* and n → π* transitions, called S_0_ → S_1_, S_0_ → S_2_ electronic conversions, which further confirms the existence of oxygen-containing functional groups in uMWCNTs [18]. The ETP-MWCNTs and ETP-uCNTs showed three absorption peaks in the UV-visible region near 230 nm, 275 nm and 350 nm, corresponding to the absorption of ETP. In addition, the ETP-MWCNTs and ETP-uCNTs showed the same energy level transition as ETP after ultraviolet absorption excitation, which confirms the ETP anchored on the surface of MWCNTs and uMWCNTs. Tus, ETP-uCNTs exhibit stronger absorption intensity due to the oxygen-containing groups providing active space for ETP anchoring, which reduces the fluorescence-quenching effect in the polymer matrix and improves the UV absorption capacity.

### 3.2. Curing Behavior of the CER-E Monomer

Most cyanate ester resin monomers are crystalline solid or semi-solid states at room temperature, and the melting point is usually around 80 °C or higher. At high temperature ( > 120 °C), the monomers transform into prepolymers with a lower viscosity [19], and the structure and cure exothermic temperatures of monomers need to be considered. The flexible molecular structure of monomers was observed in the curing process to prepare the CER materials [20]. CER-E monomer is more flexible than CER-A due to the lower steric hindrance around the core bond (-OCN), reducing steric hindrance and side chain, which decreases the final curing temperature.

The DSC thermograms for isothermal polymerization of CER-E versus time are shown in Figure 3a,b. Figure 3a shows that for the pure CER-E monomer it takes about 75 min to reach the maximum reaction rate at 180 °C, the generated exothermic flow shows that the reaction rate is low at the early stage. High temperatures provide higher reaction rates, with maximum reaction rates being reached after about 50 min at 210 °C, and at 230 °C after about 30 min. As the curing progresses, the intermediates are formed and accelerate polycyclotrimerization, and the higher temperature brings the energy to break theC≡N bond to self-polymerize, resulting in the reaction rate being increased and the time being shortened to reach the high conversion. At higher conversion rates, the viscosity of the monomer increased due to the high molecular weight, resulting in the mobility of the reactive groups being reduced and limited, and the curing reaction being controlled; the rate then drops, eventually falling to zero [21]. The CER-E/ETP-uCNT hybrid nanocomposites have similar autocatalytic behavior, but the initial conversion rate is higher than pure CER-E. In Figure 3b, it is seen that for CER-E/ETP-uCNT hybrid nanocomposites it only takes about 50 min at 150 °C, and about 30 min at 180 °C to reach the maximum reaction rate when the curing behavior reaches the same exothermic flow rate. The presence of phenol residuals and acetylacetonates of different metals can catalyze the curing reaction of cyanate ester, and interestingly the structures of ETP have similar fragments [22]. Thus, the ETP-uCNTs can catalyze CER-E polymerization and the conversion rate is higher at the initial stage.

The heat flow curves obtained from non-isothermal DSC measurements of curing behavior for CER-E monomer and the nanocomposites for the CER-E/MWCNTs, CER-E/uMWCNTs, CER-E/Eu complex and CER-E/ETP-uCNTs with 0.8 wt.% of filler loading are shown in Figure 3c. That there is a single exothermic peak on the whole curve is mainly due to the polymerization of the CER-E monomer into polycyanurate; the parameters of each composite are listed in Table 2. ΔE_a_, the apparent activation energy value of the curing reaction for CER-E hybrid nanocomposites, was calculated with Equation (1), in accordance with Ozawa’s equation [23,24]:(1)$$-\frac{\mathrm{dln}\left(\beta \right)}{d\left(\frac{1}{\mathrm{Tmax}}\right)}=1.052\frac{\mathrm{Ea}}{R}$$

The temperature of the maximum reaction rate (T_max_) was determined on the temperature position of the exothermic peak [25]. In Figure 3c, the exothermic peak temperature for the CER-E/MWCNTs was about 40 °C lower than the pure CER-E (286.2 °C), and for the CER-E/uMWCNTs it was reduced to 234.1 °C. Interestingly, a single ETP showed a very strong catalytic effect on CER-E polymerization in the CER-E/Eu complex; the peak temperature was reduced to 223.2 °C. However, the best effect was CER-E/ ETP-uCNTs; the value of T_max_ decreased by almost 68 °C, and the ΔE_a_ was also reduced from 95.7 kJ·mol^−1^ to 88.7 kJ·mol^−1^ (Table 2). The decrease of T_max_ and ΔE_a_ are a catalytic effect, indicating that CER-E monomer cured is exothermic process. Particularly, ETP-uCNTs of highly active surfaces can be chemically linked to CER-E, resulting in increased curing rates for the formation of polycyanurate network.

The chemical functionality of the curing process was analyzed with FTIR, as shown in Figure 3d. The CER-E monomer has a characteristic band with maxima near 2272 cm^−1^ and 2236 cm^−1^, relating to the stretching vibration of the cyanate group (O−C≡N) (Figure 3d). During the polycyclotrimerization, the intensity of O−C≡N groups decreases and then disappears completely at 210 °C, and the new peaks appear in the vicinity of 1367 cm^−1^ and 1564 cm^−1^, corresponding to the stretching vibrations of the cyanurate cycle structure (C=N−C, O−C−N) formed [21]. FTIR studies show the extent of the reaction for a linear relationship between the consumption of the cyanate group and the formation of the triazine ring. For the CER-E/ETP-uCNT nanocomposites, the same polycyanurate formation as pure CER-E, it is worth noting that the catalysis of ETP-uCNTs enables maximum conversion at lower curing end temperatures.

### 3.3. Thermal Stability of the CER-E-Based Hybrid Nanocomposites

The thermal stability of CER-E hybrid nanocomposites was studied using TGA analysis and the data obtained are presented in Figure 4 and Table 3. The improvement of thermal stability of composites is related to the inherent high thermal stability of organic and inorganic components, and also to the degree of interaction between the polymer matrix and the inorganic phase [26]. The addition of MWCNTs improves the thermal stability of CER-E hybrid nanocomposites due to the excellent thermal stability of the inorganic MWCNTs [25], and uMWCNTs offer the better effect because of the more effective dispersing in the CER-E matrix. In the CER-E/uMWCNTs, the uMWCNT forms a protective layer that prevents further oxidation within the CER-E network [27]. The ETP-uCNTs allow effective improvement of the thermal stability of CER-E; the value of T_d onset_ is increased by 54 °C, the T_d 10%_ reaches 436.0 °C and residual carbon reaches 47.1%. The thermal stability of the CER-E/ETP-uCNT hybrid nanocomposites is higher than that of the CER-E/MWCNTs, Among the CER-E-based nanocomposites, CER-E/ETP-uCNTs possess the higher thermal stability, which can make them applicable in a wider temperature range.

### 3.4. Mechanical Properties and Morphology (SEM Investigations) of the CER-Based Hybrid Nanocomposites

The mechanical property curves for tensile (a) and flexural (b) stress vs strain for CER-E-based hybrid nanocomposites are shown in Figure 5. Compared with the flexural stress of pure CER-E (82.25 MPa), the CER-E/MWCNTs (94.03 MPa), CER-E/uMWCNTs (115.82 MPa) and CER-E/Eu complex (ETP) (131.11 MPa) were, respectively, increased by 14.3%, 40.8% and 59.4%, which indicated that the uMWCNTs significantly increased the elongation (strain), which proved the unzipping process improved dispersion in the CER-E matrix. The loading effect of ETP was similar to that of uMWCNTs, which indicates that the Eu^3+^ forms coordination bonds with O or N atoms in the CER-E molecules, resulting in the flexural stress of CER-E/ETP-uCNTs (158.60 MPa) increasing by 92.8%. Compared with the value of tensile stress being 24.97 MPa for pure CER-E, the CER-E/MWCNTs, CER-E/uMWCNTs, CER-E/Eu complex (ETP) and CER-E/ETP-uCNT hybrid nanocomposites were 36.45 MPa, 41.03 MPa, 42.73 MPa and 48.59 MPa, respectively, increasing by 45.9%, 64.3%, 71.1% and 94.6%; the ETP-uCNTs shows the highest reinforcement effect; this is closely related with the excellent dispersion and surface activity in the CER-E matrix [9]. Moreover, the highly active modifier of the ETP provides the chemical bond between CER-E and ETP, and Eu^3+^ provides empty electron orbits, which prevent the nanocomposites from being further fractured in a tensile test [28]. Thus, a reinforcement effect of ETP-uCNTs is better than the individual MWCNTs and uMWCNTs; the ETP atoms preferentially form more stable coordinate bonds with the oxygen atoms on the surface of the uMWCNTs, capturing the lone electrons and the electrons of the large p bonds on the carbon ring, improving the mechanical properties [29]. The interface is changed from physical attraction to chemical bonding, and the ETP-anchored uMWCNTs are proved to be a linear fluorescent filler.

The mechanical properties of the composite were improved in strength as observed by tensile tests that also confirmed the above fracture SEM morphology in Figure 6a–e. The fracture surface of the pure CER-E is smooth; the polymer network indicated that the cured products are brittle. For the CER-E/MWCNTs (Figure 6a) and CER-E/uMWCNTs (Figure 6b) by tensile fracture, the draping and clearance indicated that the fracture of the nanocomposites was caused by MWCNTs or uMWCNTs pulled out from the CER-E matrix. Under the influence of an external stress, the orientation of MWCNTs are inconsistent with the loading direction of external stress, resulting in MWCNTs not participating in load transfer. MWCNTs are in large clusters and have poor mechanical properties. This behavior is the same as in Figure 5. In Figure 6b, the dispersion effect of uMWCNTS was significantly improved due to the increase of oxygen-containing functional groups on the surface. However, uMWCNTS were still incompletely bound to the CER-E matrix interface, resulting in a small gap.For the tensile fracture of the CER-E/Eu complex (ETP) (Figure 6d) shows a complex "fish scale" pattern and multi-layer dimple fracture, proving that ETP can form a chemical bond with CER-E to improve toughness. The fracture was a typical ductile fracture [30]. However, the CER-E/ETP-uCNTs show that ETP-uCNTs were well dispersed in CER-E. After the process of unzipping and ETP anchoring, the interface binding ability is stronger and the tubular structure of carbon nanotubes has no obvious pulling out, resulting in no gap between ETP-uCNTs and the CER-E matrix. ETP-uCNTs absorbs the most external stress and enhances the toughness of the nanocomposites. Fine tentacles and many microcracks appeared on the surface of the fracture, as shown in Figure 6e_2_. The strong interaction between ETP-uCNTs and CER-E matrix results in toughening effect. [31]. The results of mechanical tests demonstrated that the reinforcement effect of ETP-uCNTs was better than the individual MWCNTs, uMWCNTs and ETP.

Figure 7 shows the influence of doping concentration of MWCNTs and ETP-uCNTs on the mechanical properties of the cured CER-E/MWCNT and CER-E/ETP-uCNT nanocomposites. The MWCNTs enhance tensile strength, flexural strength and impact strength of CER-E nanocomposites. With the increase of the concentration of MWCNTs, the strength increased to the maximum at 0.8 wt.%, and an overall downward trend after exceeding this range. High concentration will lead to the aggregation of nanoparticles, resulting in weakened interaction and unable to enhance the fluorescence performance. However, the formation of ETP-uCNTs apparently provides higher mechanical properties for the CER-E. Compared to pure CER-E, the tensile strength of CER-E/0.8 wt.% ETP-uCNTs can reach 48.59 MPa, which is higher by 94.6%, the flexural strength reaches 158.60 MPa, which is an increase of 92.8%, and the impact strength reaches 26.08 kJ·m^−1^, which is an increase of 101.1%. The ETP-uCNTs with a highly active surface were adsorbed in the CER-E matrix, and reacted with the cyanate groups during the curing process, enhancing the interface bonding and further improving the mechanical properties of CER-E hybrid nanocomposites. Thus, the ETP-uCNTs as fracture nodes in the crosslinked network of the CER hybrid nanocomposites, effectively transferring stress and preventing the generation of cracks.

### 3.5. Fluorescence Characterization of CER-Based Hybrid Nanocomposites

Figure 8 shows the UV absorption spectrum for the pure CER-E and for the CER-E hybrid nanocomposites filled with MWCNTs, uMWCNTs, Eu complex (ETP) and ETP-uCNTs; these nanocomposites have the obvious absorption peaks in the UV-visible region at 225–380 nm. As the concentration is too small, the main absorption peak of the CER-E hybrid composites at 300–400 nm is the same as for pure CER-E in the ultraviolet-visible region. The liquid CER-E monomer has good color purity under ultraviolet irradiation; pure CER-E develops an amber color (Figure 8f) after curing.

In Figure 8a, the CER-E/MWCNT hybrid nanocomposites have the strongest ultraviolet absorption near 320 nm among all samples. The CER-E/uMWCNTs show a main absorption centered at near 330 nm and the peaks widen. From the above analyses, the absorption spectrum of uMWCNTs should be caused by the reactive oxygen groups on the surface due to the unzipping process. The cyanate groups (-OCN) in the CER-E monomer and the carboxylic groups in uMWCNTs react to each other, and provide enhancement of the mutual miscibility, resulting in improved ultraviolet absorption ability of nanocomposites [32]. The ETP itself and its dispersion in CER-E can effectively absorb ultraviolet radiation at 280–320 nm and 320–380 nm, and transfer the energy to Eu^3+^, resulting in excellent fluorescence [33]. In addition, the CER-E/ETP-uCNT hybrid nanocomposites demonstrates a broad intense band around 340 nm caused by the effective dispersion of uMWCNTs in the liquid CER-E monomer, and the fluorescence properties of ETP-uCNTs reach the highest value and the absorption capacity.

As can be seen from Figure 8b–c, a strong fluorescence is detected in the fluorescence spectrum of the pure CER-E. A polycyanurate network forms by curing the CER-E monomer with the appearance of fluorescence excitation and emission peaks at about 296 nm and 430 nm, respectively. The cyanate monomer, soluble oligomer and triazine cycle in the cured products also exhibit the observed fluorescence emission near 430 nm [34]. The pure CER-E and the CER-E/MWCNT nanocomposites show similar excitation and emission spectra, which proves the MWCNTs affect the curing process of CER-E, and the catalytic effect of MWCNTs on CER-E polymerization. The CER-E/Eu complex and CER-E/ETP-uCNT hybrid nanocomposites are excited at 340/374 nm by the strongest emission peaks, which could be attributed to ^5^D_0_→^7^F_J_ (J= 0–4) transitions [35]. Figure 8f shows red fluorescence upon radiation in UV light, which is attributed to the strongest ^5^D_0_→^7^F_2_ transition of Eu^3+^ centered at 612 nm. Accordingly, CER-E/ETP-uCNT hybrid nanocomposites are characterized by the highest intensity of emission fluorescence caused by the ETP-uCNTs enhancing the ultraviolet absorption ability and preventing fluorescence quenching, thus inhibiting the conjugation effect caused by aromatic accumulation of the CER-E emission peak near 430 nm; in addition, the ETP-uCNTs reduce the energy dissipation of non-radiative transitions in CER-E/ETP-uCNT hybrid nanocomposites under the condition of complete curing of CER-E [36].

Figure 8d,e show that the emission intensity of CER-based fluorescent composites tends to be stable at 430 nm as the curing reaction progresses. Compared with the CER-A monomer, the ^5^D_0_→^7^F_2_ transition was enhanced and the intensity of the conjugation of the aromatics was reduced However, the intensity of fluorescence changes with the final curing temperature, which provides a potential application as a temperature sensor for the use of the specific CER-based hybrid nanocomposites. It was detected that the fluorescence intensity decreased significantly at about 210 °C because the high-temperature environment in the curing process accelerated the attenuation of the vibration mode, thus weakening the fluorescence signal.

## 4. Conclusions

The uMWCNTs are potential linear fluorescent nanomaterials, and anchored by ETP to prepare visible fluorescence hybrid nanoparticles ETP-uCNTs.. The uMWCNTs and ETP provided structural scaffolds through mutual grafting. The ETP-uCNTs were effectively dispersed in the CER-E monomer with excellent pure chroma, highlighting that ETP-uCNTs exhibit efficient interface binding and catalytic performance. The conversion of the CER-E monomer reached a maximum at a lower curing temperature, curing to form high-density CER-E hybrid nanocomposites.

The tensile strength, flexural strength and impact strength of the CER-E/ETP-uCNT hybrid nanocomposites are improved compared with pure CER-E. Under the high concentration, ETP-uCNTs are evenly dispersed, with the maximum growth rate of strength being increased with the addition of nanofillers up to 0.8–1.0 wt.% and then remaining flat but still high.

Fluorescence properties of CER-E/ETP-uCNT hybrid nanocomposites suppressed the m-conjugated aromatic ring of the CER-E. Strong red fluorescence emission was observed under UV excitation; strong UV absorption could be applied to UV protective materials and fluorescence detection. The fluorescence intensity changes with the temperature of the curing process. The new material changes from orange resin fluorescence to special red fluorescence under uv excitation, and then to reflective blue fluorescence after thermal quenching. Fluorescence changes can be applied at high temperatures and under ultraviolet radiation for specific identification purposes.

Finally, the new CER-E/ETP-uCNTs nanocomposites can be used in the aerospace field. The excellent mechanical properties and fluorescence properties at high temperature and is used as UV protective coating. 

## Data Availability

Not applicable.

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
