# Peer review of "The Created Excellent Thermal, Mechanical and Fluorescent Properties by Doping Eu3+-Complex-Anchored Carbon Nanotubes in Polycyanate Resins"

_nanomaterials, 2022, doi:10.3390/nano12122040_

Round 1
Reviewer 1 Report
The article entitled The Created Excellent Thermal, Mechanical and Fluorescent Properties by Doping Eu3+-Complex-Anchored Carbon Nanotubes in Polycyanate Resins describes the preparation of treated MWCNTs and their utilization in composite material.
In my opinion, the article is written in a classic structure, the experiments and results are described well. Before the publication itself, I recommend that the authors should add a paragraph to the introduction and conclusion describing the benefit of the work and also with a description of the potential use of their new materials.
Author Response
Thank you very much for this valuable comment.
Please see the attachment.

Reviewer 2 Report
see attached file

Author Response

(The authors gave the same response as above.)
